# A phase I study of an adenoviral vector delivering a MUC1/CD40-ligand fusion protein in patients with advanced adenocarcinoma

Cancer vaccines as immunotherapy for solid tumours are currently in development with promising results. We report a phase 1 study of Ad-sig-hMUC1/ecdCD40L (NCT02140996), an adenoviral-vector vaccine encoding the tumour-associated antigen MUC1 linked to CD40 ligand, in patients with advanced adenocarcinoma. The primary objective of this study is safety and tolerability. We also study the immunome in vaccinated patients as a secondary outcome. This trial, while not designed to determine clinical efficacy, reports an exploratory endpoint of overall response rate. The study meets its pre-specified primary endpoint demonstrating safety and tolerability in a cohort of 21 patients with advanced adenocarcinomas (breast, lung and ovary). The maximal dose of the vaccine is $1 \times 10^{11}$ viral particles, with no dose limiting toxicities. All drug related adverse events are of low grades, most commonly injection site reactions in 15 (71%) patients. Using exploratory high-dimensional analyses, we find both quantitative and relational changes in the cancer immunome after vaccination. Our data highlights the utility of high-dimensional analyses in understanding and predicting effective immunotherapy, underscoring the importance of immune competency in cancer prognosis.

Harnessing effective immunity against cancer, such as through therapeutic cancer vaccination, promises greater specificity at potentially reduced toxicities compared to other treatments. However, this approach presents unique challenges, such as the severely immunosuppressive environment in cancer patients, particularly those with metastatic disease and heavy pre-treatment, as well as the low immunogenicity of self-tumour-associated antigens (TAAs). Hence, therapeutic cancer vaccines have overall met with suboptimal clinical results despite their attractiveness. Novel approaches are needed not only in vaccine design but also in the understanding of the immunome in cancer and its modulation by such immunotherapies.

The transmembrane mucin, MUC1, is an example of a classic TAA. Tumour-associated MUC1 is ubiquitously overexpressed, hypoglycosylated, and secreted as soluble glycoforms (e.g. CA 15-3 or CA27.29) by epithelial adenocarcinomas, including those of the breast, ovary, colon, and lung[1–3]. Structural differences between MUC1 isoforms associated with normal and cancer cells make it a highly considered target for cancer vaccination. MUC1 ranks amongst the top 75 cancer antigens prioritised by the US National Cancer Institute for translational research[2]. Despite this, multiple different MUC1 vaccines have been tested in early-phase trials with limited success[4]. There remains an unmet need and an importance in developing novel MUC1 vaccine approaches to activate the immune system and overcoming the immunosuppressive tumour microenvironment.

Immunostimulatory adjuvants can enhance, potentiate and entrench responses to TAAs[5]. One approach is to deliver co-stimulatory molecules alongside the antigen of interest. CD40 ligand (CD40L), normally expressed by CD4 T cells, binds CD40 receptors on antigen-presenting cells (APC) such as dendritic cells, and is indispensable for dendritic cell licensing[6,7]. CD40L also promotes Th1 responses[8] and may inhibit tumour growth or cause tumour cell death[9]. This ligand has been demonstrated to be highly immunogenic

✉e-mail: salvo@duke-nus.edu.sg; toh.han.chong@singhealth.com.sg

in vaccine platforms when conjugated to a TAA, as demonstrated in preclinical models[7,10]. The choice of vector also influences the immunogenicity of the vaccine response. Adenoviruses (AdV), such as human adenovirus 5 (Ad5), are leading candidate vectors due to their safety and ability to elicit both anti-viral and anti-tumour responses[11,12].

We combined these two strategies, using Ad-sig-hMUC1/ecdCD40L, a recombinant adenovirus vaccine comprising human MUC1 (hMUC1) antigen fused to the extracellular domain of the CD40 ligand (CD40L)[13–15]. In preclinical experiments, two subcutaneous injections of Ad-sig-hMUC1/ecdCD40L conferred resistance against engraftment of hMUC1+ cancer cell lines and induced regression of established hMUC1+ tumours in mouse models[13–15].

In this work, a first-in-human study of this combination vaccine construct, we undertook a Phase 1 trial in a cohort of 21 patients with MUC1-overexpressing adenocarcinomas. We achieve the primary outcome, of safety and tolerability at the highest dose of Ad-sig-hMUC1/ecdCD40L tested. To gain insight on the immunomodulatory effects of this vaccine and identify targets for further potentiation, we leverage high dimensional deep immunomics, demonstrating differences in the cancer immunome compared to matched healthy controls, and show that vaccination with Ad-sig-hMUC1/ecdCD40L results in quantitative changes in the frequencies of certain immune cell subsets as well as relational changes in immune networks. Our results suggest that the clinical outcome of immunotherapies against cancers may also depend on the integrity of the architecture of the immunome. When validated, our data may also provide an avenue to develop predictive testing of responses to immunotherapy. These findings provide a foundation for further development of MUC-1-targeted therapies, and a blueprint to leverage high-dimensional approaches for developing our understanding of the manipulation of the immunome during cancer immunotherapy.

## Results

### Study outline and patient characteristics

A total of 21 patients were enroled between September 2014 and November 2018 in this single-site study. Patient characteristics, tumour subtypes, and the number of lines of therapy are summarised in Table 1 and Supplementary Table 1. The median age was 60 (range 34–88). All patients had histologically proven locally recurrent or metastatic adenocarcinomas of the breast (n = 13, 62%), ovary (n = 7, 33%) or lung (n = 1, 5%) which is not amenable to curative resection and elevated MUC1 levels as determined by tumour immunohistochemistry or serum tumour marker at any time since diagnosis of cancer. The median number of lines of treatment in the advanced or metastatic setting prior to study entry was 3 (range 0–10).

The study drug, Ad-sig-hMUC1/ecdCD40L, is a non-replicative adenovirus (AdV) 5 vector vaccine. The vaccine vector includes a 20aa MUC-1 tandem repeat and extracellular domain of CD40L under a CMV promoter and utilises the human growth hormone secretory signal sequence (Sig), as shown in the vaccine vector diagram in Fig. 1a. The trial CONSORT diagram is as outlined in Fig. 1b.

### Safety, tolerability and clinical responses to vaccination

All patients were administered the vaccine and included in the safety and efficacy endpoint analysis. In the first part of the study (dose escalation), a single injection of Ad-sig-hMUC1/ecdCD40L vaccine was administered to 4 sequential cohorts comprising of three patients each up to a maximally administered dose of $1 \times 10^{11}$ viral particles. The vaccine was safe and well tolerated with no dose-limiting toxicities (DLTs) observed. Six additional patients were treated in expansion cohorts 5 and 6, which test for toxicity and efficacy of two and three successive administrations of the vaccine 7 days apart. It was intended to recruit a further six patients to cohort 7, which evaluates safety and efficacy of 5 vaccine administrations. However, the trial closed to recruitment after the recruitment of the 3rd patient to cohort 7 due to

## Table 1 | Patient demographics and clinical characteristics

| | Number of patients (%) N = 21 |
|---|---|
| **Total number of patients** | 21 (100) |
| **Age at consent** | |
| Median (IQR) | 60 (51.0, 65.0) |
| Range | 34–88 |
| **Gender** | |
| Male | 1 (4.8) |
| Female | 20 (95.2) |
| **Ethnic group** | |
| Chinese | 16 (76.2) |
| Indian | 1 (4.8) |
| Others | 4 (19.0) |
| **Primary disease site** | |
| Breast | 13 (61.9) |
| Lung | 1 (4.8) |
| Ovary | 7 (33.3) |
| **ECOG performance status** | |
| 0 | 6 (28.6) |
| 1 | 15 (71.4) |
| **Number of prior lines of treatment** | |
| Median | 3 |
| Range | 0 – 10 |
| **Baseline CA 15-3, U/ml (Lab reference value < 25.1 U/ml)** | |
| Median (IQR) | 37.2 (21.7–101.0) |
| Range | 11.6–634.0 |

slow accrual. All drug-related adverse events (AEs) were of low grades 1–2. Injection site reactions in 15 (71%) patients, fever in 2 (10%) and fatigue and rash occurred in one patient (5%) (Table 2). Three patients experienced grade 3 AEs which were deemed unrelated to the study drug (Supplementary Table 2). As an exploratory endpoint, objective tumour evaluation was conducted 2 months post last vaccination in cohorts 1-6 and on days 112 and 172 for cohort 7. Of the 17 patients who had measurable disease based on Response Evaluation Criteria in Solid Tumours (RECIST) v1.1, 10 had stable disease (48%) and 7 (33%) had progressive disease. There were no confirmed partial or complete responses. (Supplementary Fig. 1 and Supplementary Table 3).

### Antigen-specific responses to vaccination

To assess MUC-1 specific cytotoxic lymphocytes (CTL) responses, we assayed lymphocyte cytokine production by ELISpot using MUC1 peptides as stimuli (Supplementary Table 4). MUC1-specific IFNγ responses were significantly induced by vaccination (Supplementary Fig. 2a). This occurred despite pre-existing seropositivity to adenovirus (AdV) present in the majority (15/21) of patients prior to vaccination and the elevation of anti-AdV titres post-vaccination (Supplementary Fig. 2b, Eq. (1)). However, ELISpot has significant limitations, as: i) it has been reported in the literature that antigen-specific responses to cancer vaccines commonly do not correlate with clinical outcome[16–18] and ii) it does not have the sensitivity or dimensionality to take into account other broader immune responses in response to the intervention, such as antigenic cascade[19], which may lead to changes in the overall immunome of patients. This led us to investigate immune changes elicited by Ad-sig-hMUC1/ecdCD40L in a more holistic manner.

### Vaccination boosts cytotoxic CD8 T cells and antigen presenting CD14+ monocytes in the context of the dysregulated systemic cancer immunome

Cancer is a systemic disease perturbing the entire immunome[20,21]. Vaccination and other systemic immunotherapies may affect

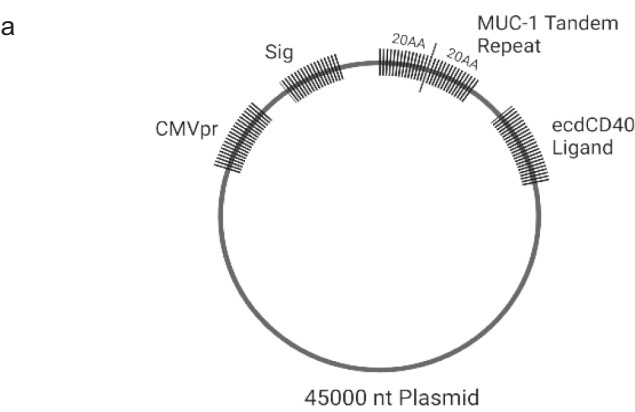

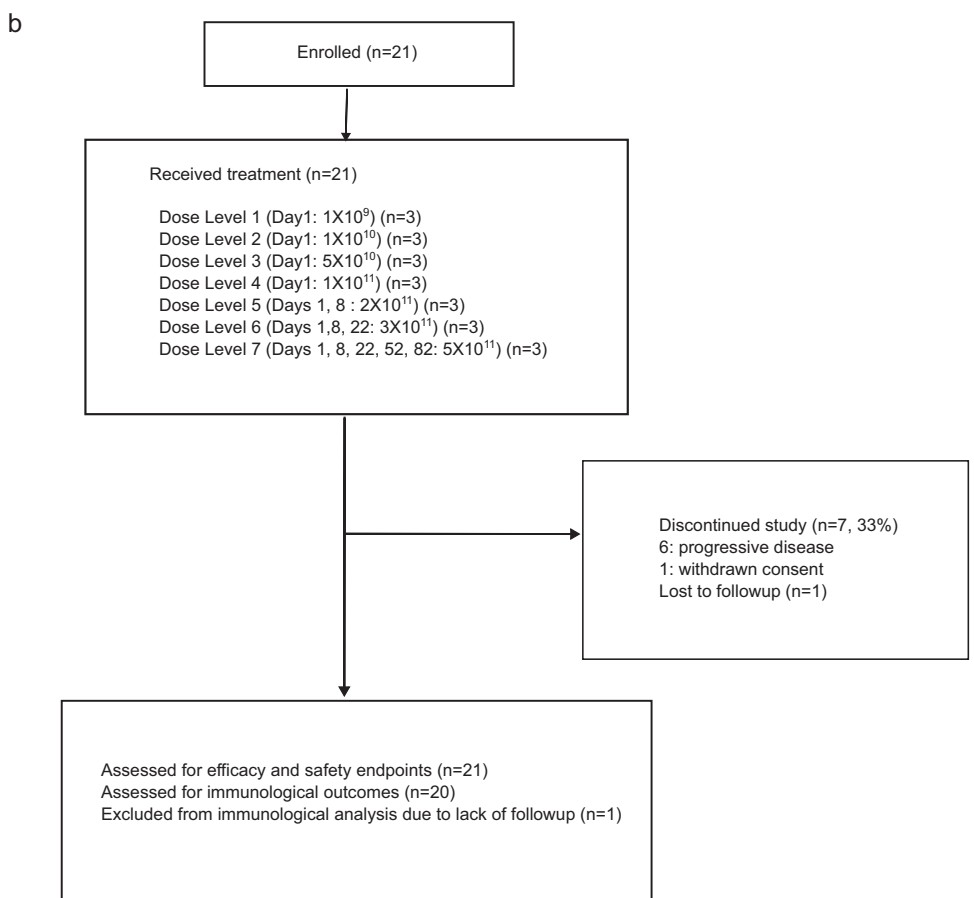

**Fig. 1 | Vaccine vector and trial flow diagram. a** Schema of vaccine expression vector. **b** Trial flow diagram and vaccination regimen. Dosage numbers indicate number of viral particles. Sig: human growth hormone signal peptide. CMVpr: cytomegalovirus promoter, ecd:extracellular domain, nt: nucleotide.

such perturbations in cancer patients. To first establish the immune mechanisms dysregulated in cancer patients compared to healthy controls, we obtained PBMCs from healthy volunteers age and sex-matched to our patient cohort[22] (see Methods). We performed cytometry by time-of-flight, (CyTOF) using two panels, Panel 1 (P1) being a T cell focused panel, and Panel 2 (P2) a B cell/APC focused panel) (Supplementary Tables 5 and 6). In total, we utilised 71 unique surface and intracellular markers. As P2 was focused on B cells and APCs, we gated and manually removed CD3 cells before analysis. Our mass cytometry data was then analysed through the web application and workflow, EPIC, a tool

that we developed to depict and dissect the human systemic immunome[23].

For an initial summarisation of the differences between baseline cancer patients and healthy controls, we performed a tSNE analysis, similar to principal component analysis (PCA), on the mass cytometry data (Supplementary Fig. 3a). The resulting tSNE graph showed that the immune architecture of cancer patients was organised distinctly from healthy individuals. Within the cancer group, a few patients clustered more closely to healthy subjects than others, suggesting a possible stratification based on heterogeneity in immune dysregulation among cancer patients. Heterogeneity did not appear to be due to

**Table 2 | Adverse events (AE) deemed possibly, probably, or certainly related to study drug graded according to Common Terminology Criteria for Adverse Events (CTCAE) 4.0**

| AE | Grade 1 | Grade 2 | Grade ≥ 3 | Any Grade |
|---|---|---|---|---|
| Injection site reaction | 9 (43%) | 6 (29%) | 0 | 15 (71%) |
| Fever | 2 (10%) | 0 | 0 | 2 (10%) |
| Fatigue | 1 (5%) | 0 | 0 | 1 (5%) |
| Rash | 1 (5%) | 0 | 0 | 1 (5%) |

Occurrence of adverse events are displayed as percentage of the 21 patients enroled in the study.

the type of cancer, which were all adenocarcinoma (Supplementary Fig. 3b). There were no significant differences in the changes in tumour measurements by dose level or cancer type (Supplementary Fig. 3c, d) hence we performed further analyses on the cancer cohort as a whole.

To further investigate the differences in immune subsets between healthy immunomes and those of cancer patients, we first visualised the distribution of cells on a tSNE map. In this type of visualisation, cells with similar expression markers are grouped together. Using data from P1, we created a reference tSNE map by overlaying manually gated major immune lineages, such that we could compare the distributions of broad immune subsets based on their location on the tSNE graph (Supplementary Fig. 4a(i)). We then compared these distributions to cell density embedded graphs projected on the same tSNE maps, across healthy, pre-vaccine cancer baseline, and post-vaccine (Supplementary Fig. 4a(ii)). On a broad level, healthy immunomes had a significantly greater proportion of CD8 + T cells, as confirmed by manual gating of broad immune subsets (Supplementary Fig. 4a(iii)). Going into further detail, we investigated specific immune differences using our analysis pipeline (Supplementary Fig. 4b), and identified 26/64 clusters significantly different between healthy and pre-vaccine cancer. Repeating the same type of analysis using P2, we observed that CD14 + monocytes were significantly decreased in pre-vaccine cancer samples compared to healthy (Supplementary Fig. 2a(i–iii)), with 13/36 clusters significantly different between healthy and pre-vaccine cancer. In total, cancer and healthy immunomes differed significantly in frequency in 39/100 clusters identified. Cluster frequencies are reported in Supplementary Figs. 6 and 7. Echoing what we observed using broad immune phenotypes, IFNγ-producing cytotoxic CD8 (P1_47, Supplementary Figs. 4b and 6g), as well as CD14 + monocytes (P2_31, Supplementary Figs. 5b and 7a), were deficient in cancer compared to healthy immunomes. This analysis also identified more subtle changes that were not reflected in changes of major immune subsets. For example, not all clusters were uniformly decreased in cancer, with a population of TBET- memory CD4 T cells (P1_28, Supplementary Figs. 4b and 6a) and CD14- CX3CR1 + monocytes (P2_25, Supplementary Figs. 5b and 7c) being elevated in cancer. This highlights the global dysregulation of the cancer immunome, with an involvement of both adaptive and innate arms of the immune system.

We then investigated the changes in the immunome of cancer patients after vaccination. One striking difference was the increase in IFNγ + and granzyme B (GZMB) expressing cytotoxic CD8s after vaccination, which could have potential anti-tumour activity (Fig. 2a). As we noticed that several IFNγ and GZMB-expressing CD8 clusters had a similar phenotype, we combined them during validation by manual gating (Fig. 2b), and confirmed that overall, these immune cells were significantly induced by vaccination. The innate immune system, specifically CD14 + monocytes and CD1c + dendritic cells (Fig. 2c), were also induced by vaccination, as determined by our unsupervised analyses. Manual gating confirmed that CD14 + monocytes were significantly increased post-vaccination (Fig. 2d). This suggests that

vaccination results in quantifiable immune shifts even in immune-dysregulated cancer patients.

## Vaccination reorganises the global immune network and increases the complexity of node connections

To represent these immune differences in a relational manner rather than by cell frequency changes, we visualised the immunome at a systems level using a network analysis framework as first described in Kumar et al.[24] Briefly, proportions of cellular subsets (nodes/clusters) from P1 and P2 mass cytometry data were calculated, using 10 samples each from pre and post-vaccination groups (Supplementary Table 7). Pairwise correlations between each node pair were then calculated to build a correlation-based immune cell network. To define the edge, or interaction, between two nodes, the absolute coefficient cut-off of 0.6 or above was used. The network was plotted using force-directed layout. The colour of these connecting edges represents the direction of correlation (blue: negative correlation, yellow: positive correlation) (Fig. 3a, pre-vaccination, and Fig. 3b, post-vaccination). We manually assigned each node to a cellular phenotype based on lineage marker expression (CD4, CD8, NK cells, B cells, monocytes, and γδT cells). Comparing the properties of network graphs of cancer patients pre- and post-vaccination, vaccinated patients increased the number of positively related connections (from 260 to 281) while decreasing negatively correlated connections (from 101 to 76). Modularity (restricted connections to between groups of nodes, or modules, rather than the wider network) also decreased from 0.118 to 0.104 (Supplementary Table 8). We then overlaid effector (GZMB, IFNγ, or TNFα) expression onto the networks (Supplementary Fig. 8). In the pre-vaccination network, we noticed that a subset of effector CD8 nodes (black asterisks, Fig. 3a, Supplementary Fig. 8a) formed a module on the edge of the network. After vaccination, these nodes became more distributed in the network, forming connections with an increased number of nodes and cell types (Fig. 3b, Supplementary Fig. 8b). This was exemplified in node P1_48, a GZMB-expressing CD8 node, which was connected to three other nodes pre-vaccination (Supplementary Fig. 9a). Post-vaccination, P1_48 had an increased number of connections, particularly to CD4 T cell nodes (Supplementary Fig. 9b). Interestingly, the connection to P1_13, a Treg node, was preserved, suggesting a possible functional relationship between the two cell types. We also observed a similar pattern in P1_40, a memory CD8 node co-expressing IFNγ and PD-1. In pre-vaccination cancer, P1_40 had few connections, mostly to other effector CD8 nodes (Fig. 3a, black asterisks, Supplementary Fig. 9c). Vaccination increased the number and variety of correlated nodes (Supplementary Fig. 9d). Interestingly, after vaccination, P1_40 formed a module with other nodes expressing inhibitory markers such as CTLA-4 (P1_7, P1_1, P1_16 and P1_9) and Tregs (P1_8). Altogether, these data suggest that vaccination can, in a limited fashion, re-invigorate the deficient immune systems of cancer patients in a relational manner.

## Elevation of GZMB + CD8 T cells and B cells after vaccination in stable disease patients

Inherent heterogeneity in immunological responses could explain divergent outcomes to vaccine therapy. To investigate immune responses associated with better clinical outcome, we examined both clinical and immunological data for possible stratification of responses. Based on clinical RECIST v.1.1 classification[25], patients could be divided into those with stable disease (SD) and those with progressive disease (PD) (Supplementary Table 3). There were 4 patients that could not be classified by RECIST, A01, A06, A08 and A20. These patients were excluded from stratified analyses.

We then investigated the immune response to vaccination in SD patients. Immune cell subsets in SD patients before and after vaccination were compared for significant differences using Mann–Whitney U test. Unsupervised analyses identified cytotoxic CD8 T cells

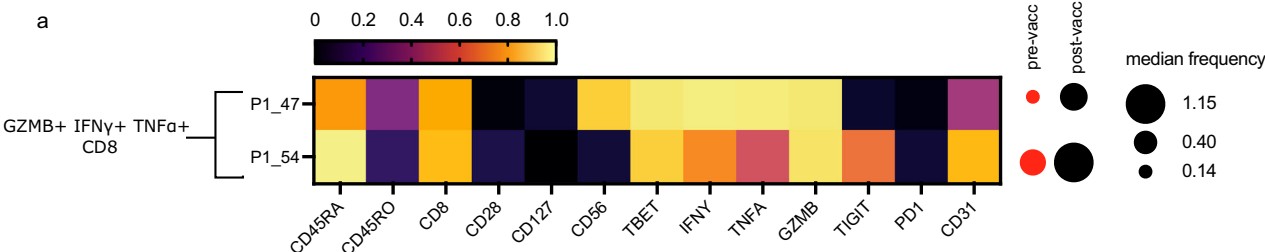

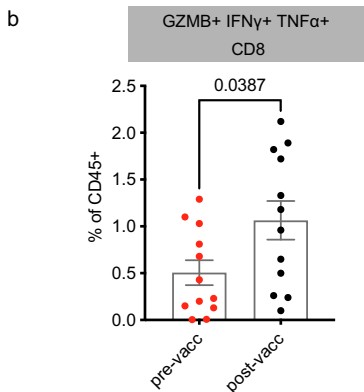

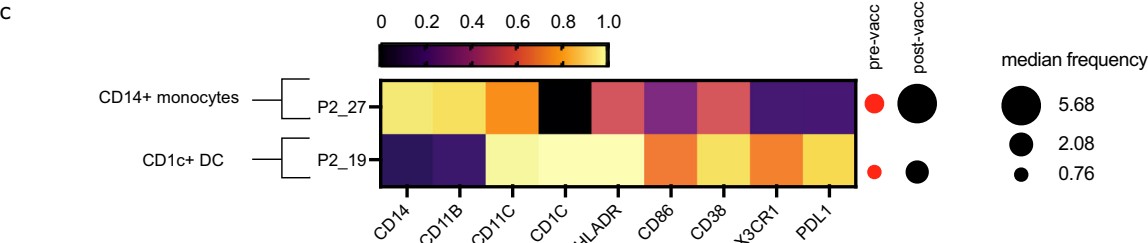

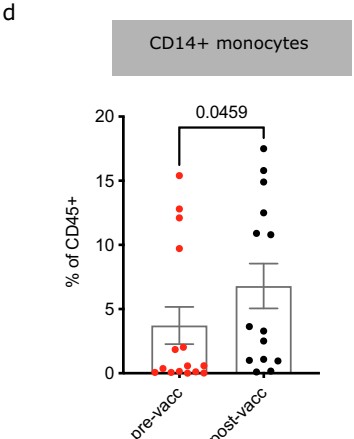

**Fig. 2 | Vaccination is associated with increases in cytotoxic CD8 T cells and antigen presenting monocytes.** Mass cytometry data was clustered by FlowSOM and analysed for clusters that differed pre (red) and post (black)-vaccination. **a** Cluster phenotypes identified using a T cell focused panel (P1). **b** Manually gated cytotoxic CD8 T cells. Pre-vaccination: $n = 12$ patients, post-vaccination: $n = 12$ **c** Cluster phenotypes identified using a B cell/APC focused panel (P2). **d** Manually gated clusters in (**c**) Pre-vaccination: $n = 15$, post-vaccination: $n = 14$. Heatmap represents the scaled arcsin median expression of each marker. *P*-values were computed by Mann–Whitney U test (two-tailed). Error bars indicate SE of mean. For each staining panel, some samples were excluded from supervised analyses due to insufficient cell numbers. Source data are provided as a Source Data file.

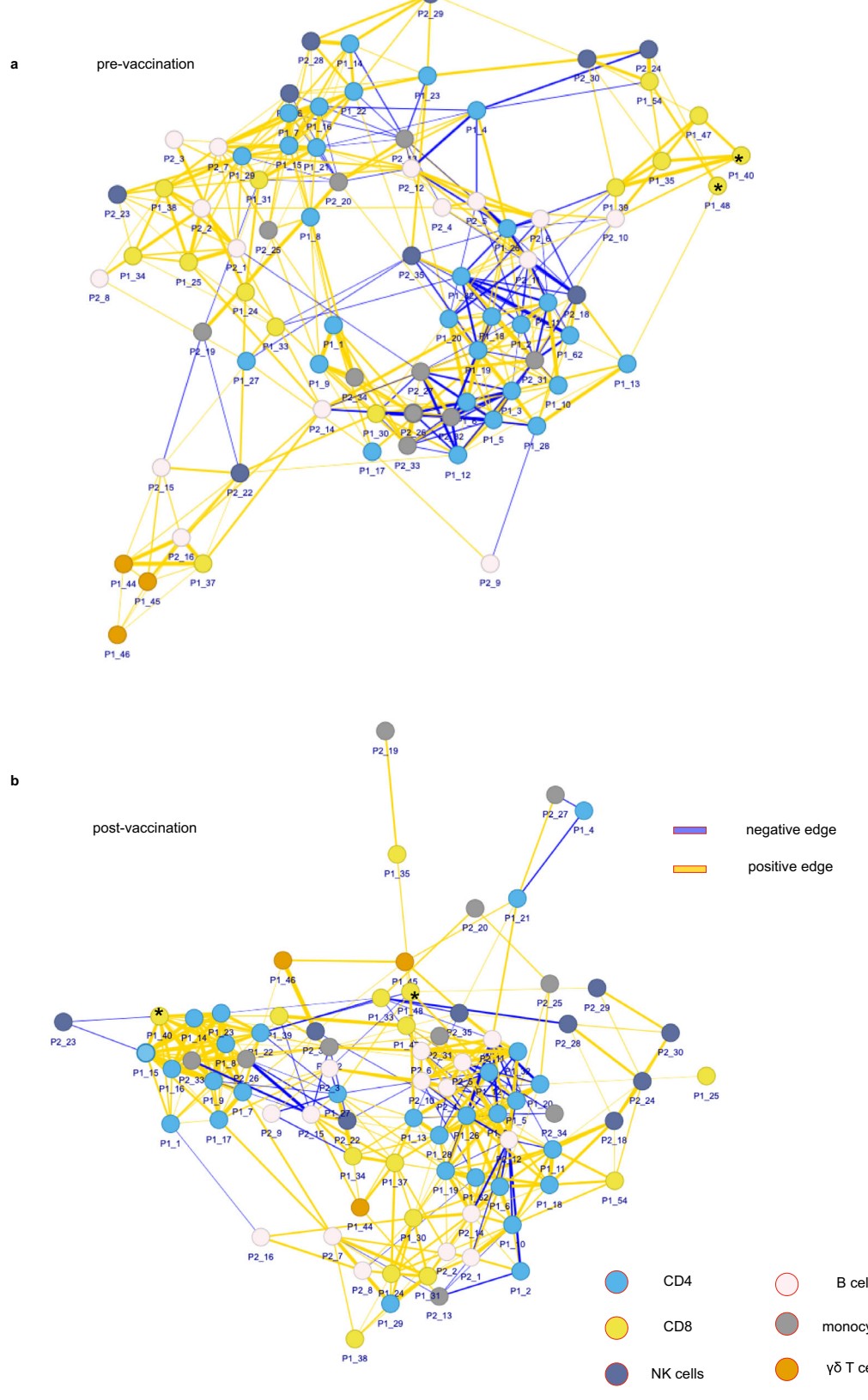

**Fig. 3 | Vaccination reshapes immune networks in cancer patients.** Network analysis of mass cytometry data in *n* = 10 patients (**a**) pre and (**b**) post-vaccination (correlation coefficient: 0.6). Yellow edges represent negative correlations and blue edges represent positive correlations. Nodes are coloured by cell type (blue: CD4, yellow: CD8, beige: B cells, indigo: NK cells, grey: monocytes, orange: γδ T cells). The thickness of the line represents the strength of the correlation. Asterisks: nodes of interest. Source data are provided as a Source Data file.

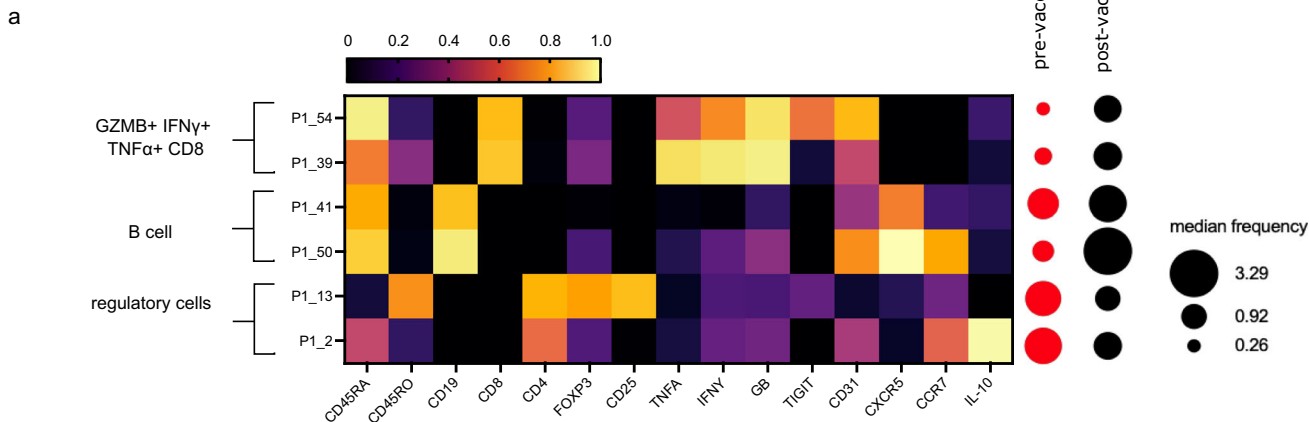

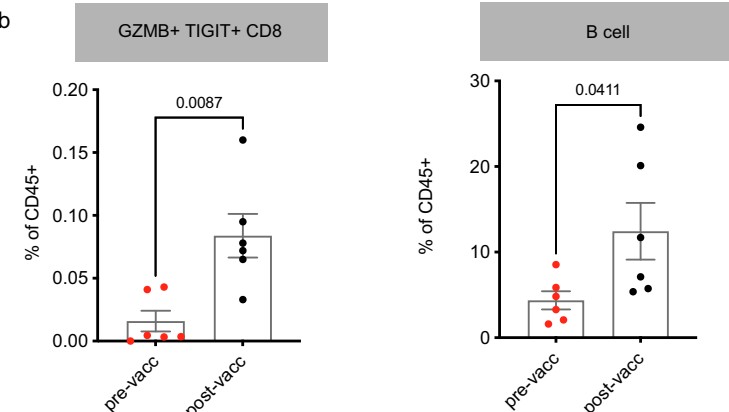

**Fig. 4 | Stable disease patients show increased cytotoxic CD8s and B cells post-vaccination. a** Mass cytometry data were clustered by FlowSOM and analysed for clusters that differed pre and post-vaccination in patients with stable disease as defined by RECIST v.1.1. The phenotypes of these clusters are summarised in the heatmaps of marker expression. The median frequency of each cluster was represented by the size of the dots. **b** Manual gating validation of selected subsets in

(**a**) Heatmap represents the scaled arcsin median expression of each marker. Mann–Whitney U tests (two-tailed) were performed to compare between groups (n = 6 patients). Error bars indicate SE of the mean. Some samples were excluded from supervised analyses due to insufficient cell numbers. Source data are provided as a Source Data file.

expressing IFNγ and GZMB, as well as CXCR5 + B cells (Fig. 4a), as clusters that increased in frequency after vaccination. We verified the increase of cytotoxic CD8 T cells and CXCR5 + B cells by manual gating (Fig. 4b). Interestingly, these were subsets that were significantly different between healthy controls and cancer patients at baseline (Supplementary Fig. 6g). There was also a slight decrease in regulatory FOXP3 + and IL-10 + CD4 T cells, but this was not significant when verified by supervised gating. We also compared patients with progressive disease (PD) pre- and post-vaccination (Supplementary Fig. 10). However, the sample size was too small to achieve significance.

This suggests that vaccination with Ad-sig-hMUC1/ecdCD40L elicits contrasting immune responses in SD patients compared to those with PD. This contrast in the architecture of the immunome is likely the cornerstone of the clinical fate of the intervention.

## Discussion

We report here a first-in-human, phase 1 study of Ad-sig-hMUC1/ecdCD40L therapeutic cancer vaccine directed against the MUC1 tumour antigen. The trial achieved the primary endpoints of safety and tolerability, with no grade 3 or higher drug-related AEs reported even at the highest dose tested (Table 2). This encouraging safety profile

paves the way for further studies in larger cohorts to determine the optimum dosage for this vaccine.

Without claiming, as obvious for this stage of development, clinical efficacy, and despite the heavily pre-treated, advanced, and often widely metastatic disease of the patients enroled (Table 1), we were able to separate patients based on the presence of clinical benefit, based on validated criteria[25]. Importantly, our analysis showed that the baseline immunomes of the cancer patients in our study were broadly dysregulated compared with healthy controls, with significant deviations in 39 out of 100 immune cell clusters found in our high-dimensional analysis (Supplementary Figs. 4 and 5). In particular, there were notable differences in cytotoxic IFNγ + GZMB + CD8, and CD14 cell frequencies (Supplementary Figs. 4b, 5b, 6g and 7a), highlighting the dysfunction of both adaptive and innate immunity. Despite this, vaccination had quantitative and qualitative effects on the immune system. Vaccination increased the proportion of cytotoxic CD8 T cells (Fig. 2b) as well as CD14 + monocytes (Fig. 2d) which were significantly different between healthy and baseline cancer (Supplementary Figs. 6g and 7a).

Using network analysis, we demonstrated that vaccination partially restored the connectivity of immune networks in cancer. Cancer

immunomes pre-vaccination had higher modularity (restricted connections, which was reduced by vaccination (Fig. 3, Supplementary Table 8). This pattern was particularly pronounced when looking at individual nodes (P1_48, Supplementary Fig. 9a, b and P1_40, Supplementary Fig. 9c, d), where pre-vaccination patients had markedly reduced connectivity and complexity of networks compared to post-vaccination. This suggests that in cancer, specific immune cell types may have a limited ability to interact with, influence and/or be influenced by other cell types thus compromising effective anti-tumour immunosurveillance. Connectivity appears to be partially recovered after vaccination. In particular for PD-1+ nodes such as P1_40, future studies could determine whether this sub-population represents a pre-exhaustion, but still functional, subset of T cells[26–28] and if future combinations of this therapy, such as with anti-PD-1 immune checkpoint blockade, for example, could be explored.

This pattern of immune restoration was specifically present in a subset of our patients cohort who had SD disease by RECIST classification. SD patients showed potentially beneficial significant increases in cytotoxic CD8 T cells and B cells (Fig. 4a, b), echoing the overall changes in response to vaccination (Fig. 2b, d).

Our data suggests a broad effect of vaccination activating multiple arms of the immune system, including potentially oncolytic CD8s (Fig. 2b), an expected outcome from an adenoviral vector[11,29]. Vaccination also increased frequencies of monocytic antigen-presenting cells (CD14 + cells, P2_27, Fig. 2d), possibly activated by CD40L, which is broadly immunostimulatory[14,30], and has been shown to improve immunogenicity in multiple pre-clinical and clinical studies[10,30,31]. These data corroborate accumulating evidence that an efficient immune system is necessary systemically to effectively counteract cancer[32] and that systemic immunotherapies must successfully modulate not only the cancer microenvironment but also the peripheral immune system. Our findings, as well as our previous experience in multiple disease settings[24,33–35], demonstrate that high-dimensional methods can identify subtle changes in response to therapies that can be targeted and enhanced in further studies.

In summary, we have demonstrated that Ad-sig-hMUC1/ecdCD40L vaccine is safe and tolerable, supporting further testing of this vaccine in a larger cohort. A follow up study combining this vaccine with an immune checkpoint inhibitor is planned in advanced adenocarcinoma patients, and we believe that its design will benefit from our data and approach. Our study has several limitations. Given the pilot nature of this Phase I trial and small number of patients, our clinical results are not conclusive. However, our systematic, high-dimensional approach provided a unique opportunity to study the immune mechanisms of the immunotherapy tested. Importantly, we have identified cancer-related aberrations in the architecture and function of the immunome, and which components of immune competency are restored by cancer vaccination in a clinically meaningful and potentially predictive fashion. This will allow for more focused scientific questions in future clinical studies, such as exploring synergistic therapies for augmenting the immunome shifts identified by our analyses.

## Methods
### Study design
All research in this open-label phase 1 trial (ClinicalTrials.gov identifier: NCT02140996) was approved by the SingHealth centralised institutional review board. Written informed consent was obtained from all patients before enrolment. All procedures involving human participants were carried out in accordance with the Declaration of Helsinki and principles of Good Clinical Practice. Patients aged 21 years or older with histologically proven locally recurrent or metastatic adenocarcinoma of the breast, ovary, lung, colon or prostate not amenable to curative surgery were eligible. Other key eligibility for inclusion include measurable or non-measurable disease, elevated MUC-1 levels

as measured by elevated serum tumour marker CA 15-3 or CA 27.29 at any time since diagnosis of cancer, ECOG performance status 0-2 with life expectancy of greater than 12 months, satisfactory organ function and receipt of at least 1 prior line of palliative chemotherapy. Patients were excluded if they had central nervous system disease, ornithine transcarbamylase deficiency, any history of bronchospasm or asthma requiring steroid therapy, any autoimmune diseases, prior organ transplant or any other malignancies. Concurrent steroid or anti-cancer therapy including tamoxifen was not permitted within 28 days of trial registration. Patients were recruited from September 2014 to November 2018. Of note, aromatase inhibitors and raloxifene were allowed to be administered concurrently with study vaccine. The study comprised 3 parts, a dose-escalation and 3 dose-expansion cohorts evaluating safety and efficacy of 2, 3 and 5 repeated vaccine administration. The primary endpoint of the dose-escalation phase was to establish the maximum tolerable dose (MTD), safety and tolerability of the vaccine. The secondary endpoint was to determine the optimal biological dose and schedule associated with an immune response as assessed by immunological profiling and clinical activity. Adverse events (AE) were graded using the National Cancer Institute Common Terminology Criteria for Adverse Events (CTCAE v.4.0).

The dose-escalation phase of the trial utilised a 3 + 3 design (summarised in Supplementary Fig. 1). We enroled patients in cohorts of three for up to four dose levels (single subcutaneous administration of $1 \times 10^9$, $1 \times 10^{10}$, $5 \times 10^{10}$, $1 \times 10^{11}$ viral particles) to establish the MTD. Patients in subsequent three cohorts were treated with multiple subcutaneous injections at the MTD, or if one is not established, the highest dose level stipulated in our protocol (i.e., $1 \times 10^{11}$ viral particles). Patients in cohort 5 received vaccinations on days 1 and 8, and patients in cohort 6 days received the vaccine on days 1, 8, and 22. If no dose-limiting toxicity (DLT) is observed in cohort 6, then a further six patients were enroled into cohort 7 (the dose-expansion phase of the trial), in which patients received injections on days 1, 8, 22, 52 and 82. DLT was defined as any treatment-related grade 3 or greater AEs, or grade 2 or greater allergic or immune-mediated reactions, within the first 28 days after vaccination. If one patient developed a DLT, we would include another three patients at that dose level, and dose escalation would continue if no additional DLTs occurred. If two patients developed DLTs, we would define the preceding dose level as the MTD. Further information is available in the Study Protocol (Supplementary Information).

### Clinical procedures
The Ad-sig-hMUC1/ecdCD40L vaccine was manufactured at an US Food and Drug Administration-approved facility with expertise in adenoviral vaccine production according to Good Manufacturing Practice guidelines. We evaluated patients for treatment AEs using physical examination, laboratory tests of serum biochemistry and complete blood counts, and vital signs, including an electro-cardiogram at regular predetermined time points. Peripheral blood samples were obtained for human leucocyte antigen (HLA) typing, monitoring of CA 15-3, and other immune biomarkers. Tumour restaging scans were performed 2 months after the last vaccination for cohorts 1-6 and days 112 and 172 in cohort 7. For peripheral blood mononuclear cells (PBMC) and sera collection, up to 40 mL of peripheral blood were drawn from consented patients at baseline pre-vaccination and stipulated post-vaccination time points. Patients' peripheral blood samples were diluted 1:2 with DPBS, followed by isolation of PMBC using standard Ficoll density gradient centrifugation techniques. The isolated PBMCs and plasma samples were cryopreserved until use.

### Clinical data analysis
All patients who met the eligibility criteria and received at least one dose of the study vaccine were included in safety and efficacy analyses.

Patient demographics and baseline clinical characteristics were tabulated and summarised using descriptive statistics. If a patient experienced more than one incident of the same adverse event during the trial, the worst grade experienced by the patient was reported. DLT experienced by patients were listed by dose level and the worse grade experience by each patient within the first 28 days tabulated. Overall response was evaluated according to the Response Evaluation Criteria in Solid Tumours (RECIST v.1.1)[25] for patients with the measurable disease to define the stable and progressive disease. Following completion of protocol-defined radiological assessments at month 2 for cohorts 1-6 and day 172 for cohort 7, patients continued usual oncological management as per routine care. All analyses were performed using SAS version 9.4 (SAS Institute Inc.) and $p < 0.05$ was considered to denote statistical significance.

### Healthy human donors

Peripheral blood was collected from age and sex-matched healthy human donors with informed consent at KK Women's and Children's Hospital as described[22]. Ethical approval was given by the SingHealth Centralised Institutional Review Board.

### MUC1 peptides and enzyme-linked immunospot (ELISpot) assay

A peptide pool that contains six 15 amino-acid long peptides with 11 overlapping amino-acid sequences and 4 offset amino acid sequences that span the sequence of two MUC1 extracellular domain tandem repeats were synthesised (JPT Peptide Technologies, Berlin, Germany; Supplementary Table 4). Peptides were dissolved in DMSO and mixed into a 40 μg/mL/peptide stock then stored at −80 °C until use.

Interferon (IFN)γ ELISpot assays were performed where PBMCs were available as described by the manufacturer (Human IFNγ ELISpot^PLUS, Mabtech AB, Nacka, Sweden). Briefly, 96-well plates pre-coated with capture antibody against interferon IFNγ were washed with phosphate-buffered saline (PBS) and blocked with culture media (45% RPMI 1640 / 45% EHAA + 10% FCS and 2mM L-glutamine) for at least 30 minutes. $3 \times 10^5$ thawed PBMCs per well were then incubated with the MUC1 peptide pool at 1 μg/mL/peptide with 2.5% v/v DMSO content. Anti-CD3 antibody supplied in the ELISpot kits was used as positive-control and culture media containing 2.5% v/v DMSO as negative-controls. After 16 to 20 hours, the plates were developed according to manufacturer's protocol and analysed using an Immunospot analyzer (CTL Analyzers, USA). Spot-forming cells (SFCs) frequencies were calculated and expressed as SFC per $10^5$ cells. Background values were defined as the mean numbers of the negative controls and deducted from the mean numbers obtained with the MUC1 peptide pool. Mean spot counts of >2 x background values were considered positive.

### Measurement of plasma anti-adenovirus antibody

Qualitative detection of IgG antibodies against adenovirus antigens in the plasma was performed by ELISA assay as described by the manufacturer (Anti-Adenovirus IgG Human ELISA Kit, Abcam, UK). 96-well plates pre-coated with adenovirus antigens were used. The cryopreserved plasma from 1:2 diluted blood samples were thawed, diluted 1:50 with sample diluent and added to the plates at 100 μL/well, then incubated for 1 hour at 37 °C. Bound antibodies were detected by a second incubation with anti-human antibodies conjugated with horseradish peroxidase. After adding substrate and stop solution, the change in colour to yellow indicated positive samples. The absorbance was determined for each well at 450 nm and reference wavelength at 620 nm. Results in Standard Units were calculated by:

$$\frac{\text{Patient (mean) absorbance} \times 10}{\text{Mean Cut} - \text{off control absorbance}} = \text{Standard Units} \quad (1)$$

Standard Unit values of >11 were considered positive, <9 were considered negative, and 9-11 were considered inconclusive.

### Statistics and reproducibility

Sample size was determined by study recruitment. PBMC samples were processed and acquired for mass cytometry blind. Samples were unblinded after de-barcoding and prior to statistical analysis. Patient A08 was excluded from immunological analyses as they were lost to follow-up. Additionally, samples with too few viable cells were excluded during QC steps of mass cytometry data analysis. The marker 167Er-TCRVα7.2 was excluded from clustering analyses in our routine quality control steps due to signal spillover from 168Er-IFNγ. Replication was not possible due to the restricted number of available samples. Reproducibility between mass cytometry runs was ensured with the use of a bridging control.

### Mass cytometry and data analysis

Thawed PBMC were rested for 30 minutes at 37°C before being divided for further manipulation as previously described. To induce cytokine production, cells were stimulated with 150 ng/mL phorbol 12-myristate 13 acetate (PMA) and 250 ng/mL ionomycin (Sigma-Aldrich) for 4 hours, with Brefeldin A and monensin (eBioscience) added after the first hour of incubation. Cells were then stained with two separate panels of metal-conjugated antibodies as listed in Supplementary Tables 5 and 6. After staining and fixation with P1 or P2, cells were stored in 1% paraformaldehyde overnight or until acquisition. Cells were acquired with a Helios mass cytometer (Fluidigm). Data were then analysed as previously described. Briefly, live singlet cells were de-barcoded, and samples containing less than 5000 live cells excluded from further analysis. CyTOF data was batch normalised and subjected to unsupervised clustering using the self-organising map (SOM) algorithm[36] to group cells with similar expression patterns into immune cell subsets. Clusters displaying statistically significant frequency differences between patient groups were identified by Kruskal–Wallis multi-group test or Mann–Whitney U pair-wise rank-sum tests. The high-dimensional immune landscape was visualised on 2-dimensional t-SNE coordinates. Fast-interpolation (fit-SNE)[37] implementation was used for t-SNE analysis. In some cases, phenotypically similar clusters were combined for analyses. Supervised manual gating was used to validate unsupervised analyses and exclude artefacts. Samples were excluded from analyses if there were insufficient cells for gating. Gating strategies are illustrated in Supplementary Fig. 11. For comparisons between groups, "post-vaccination" was defined as at least 14 days after a vaccination dose. Statistical analyses and graphing were performed using R statistical software, R packages, and GraphPad Prism. Appropriate statistical tests were performed as described in figure legends, with results being considered statistically significant if $p < 0.05$. The pipeline used for mass cytometry analysis can be accessed at https://epicimmuneatlas.org.

### Network analysis

Network analysis was performed as described[24]. Briefly, node frequencies were calculated for each patient and the pairwise correlation between each node calculated. To construct the network, nodes were connected by specifying absolute correlation coefficients as shown in the figures and figure legends. Networks were plotted using force-directed layout. Various properties of the networks were calculated using the igraph R package. The network diagrams were plotted using the visNetwork R package.

### Reporting summary

Further information on research design is available in the Nature Research Reporting Summary linked to this article.

## Data availability

Data availability is subject to local rules and regulations. Subjects did not provide consent for their data to be made publicly available. However, every reasonable effort will be made to promptly satisfy scientifically valid requests. Requests for data should be made to the corresponding authors together with a detailed study plan and a commitment not to use the data and its derivatives for commercial purposes. The proposal will require approval by the SingHealth Centralised Institutional Review Board, National Cancer Centre Singapore (NCCS), Singapore Clinical Research Institute (SCRI), and the Principal Investigators of the study. Requesting researchers will be required to sign a data access agreement with the relevant parties. Upon signing, a mini dataset of de-identified raw data relevant to the results shown in the manuscript will be uploaded to a password protected account. Patient-related data were generated as part of a clinical trial and are subject to patient confidentiality. Access to de-identified clinical data, raw ELIspot and ELISA assay data can be made available for a period of 2 years. The study protocol and statistical analysis plan are available in the Supplementary Information file. The remaining data are available within the Article, Supplementary Information, or Source Data file. Source data are provided with this paper.

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

## Acknowledgements

Support for this study was provided by a grant from the National Medical Research Council Singapore (NMRC/1305/2011) awarded to H.C.T. MicroVAX, LLC provided study vaccine and contributed funds for study operations. Singapore Clinical Research Institute provided study sponsorship. MicroVAX, LLC were not involved in study design, data collection, analysis, or manuscript writing apart from review of manuscript for accuracy prior to submission for publication. We acknowledge investigators Daniel S.W. Tan, Matthew Ng, David Tai, Justina Lam and all site staff at the National Cancer Centre Singapore for their support in the conduct of this study. Finally, we would like to thank the patients who participated in this study.

## Author contributions

Conceptualisation: H.C.T. and S.A.; Methodology: H.C.T., J.W-K.C., S.H.T., X.L., A.O.L.S.; Formal Analysis: P.K., M.W., and S-H.T.; Investigation: T.J.T., W.X.G.A., W-W.W., J. W-K. C., R.C., H-S.C., J.W-K.C., X.L., W.H.S., R.B., N.K., A.C.T., A.S., B.A., J.E.C., A.M.L., A.O.L.S., R.H., D.G., and H.C.T.; Resources: T.J.T., W.X.G.A, W-W.W, T.A., J.W-K.C, H-S.C., A.C.T., A.O.L.S., S.A., H.C.T.; Data Curation: T.J.T., H-S.C., S.H.T., X.L.; Writing — Original Draft: H.C.T., N.L.S., S.A., T.J.T., W.X.G.A. and W-W.W.; Writing— Review and Editing: T.J.T., W.X.G.A., W-W.W., H-S.C., S-H.T., R.C., J. W-K.C., N.L.S., W.H.S., R.B., N.K., B.A., R.H., X.L., A.C.T., A.O.L.S., J.E.C., V.C., A.M.L., D.G., M.Z.W.C., M.W., P.K., S.A., H.C.T.; Validation: all authors; Visualisation: T.J.T., W.X.G.A., W-W.W., S-H.T., W.H.S. and R.H.; Software: P.K., M.W., S-H. T. and X.L.; Supervision: H.C.T. and S.A.; Project Administration: X.L., R.C. and H-S.C.; Funding Acquisition: H.C.T., J.W-K.C, S.A.

## Competing interests

The authors have no competing interests to declare.

## Additional information

**Tira J. Tan**[1,9], **W. X. Gladys Ang** [2,9], **Who-Whong Wang**[1,9], **Hui-Shan Chong**[3], **Sze Huey Tan** [3], **Rachael Cheong**[1], **John Whay-Kuang Chia**[4], **Nicholas L. Syn**[5], **Wai Ho Shuen** [1], **Rebecca Ba**[1], **Nivashini Kaliaperumal**[6], **Bijin Au**[6], **Richard Hopkins**[6], **Xinhua Li**[7], **Aaron C. Tan** [1], **Amanda O. L. Seet**[1], **John E. Connolly**[6], **Thaschawee Arkachaisri** [8], **Valerie Chew** [2], **Ahmad bin Mohamed Lajam**[2], **Dianyan Guo**[2], **Marvin Z. W. Chew**[2], **Martin Wasser** [2], **Pavanish Kumar**[2], **Salvatore Albani** [2,10] ✉ **& Han Chong Toh** [1,10] ✉

[1]Division of Medical Oncology, National Cancer Centre, Singapore, Singapore. [2]Translational Immunology Institute, SingHealth Duke-NUS Academic Medical Centre, Singapore, Singapore. [3]Division of Clinical Trials and Epidemiological Sciences, National Cancer Centre, Singapore, Singapore. [4]Curie Oncology, Singapore, Singapore. [5]Yong Loo Lin School of Medicine, National University of Singapore, Singapore, Singapore. [6]Institute of Molecular and Cellular Biology (IMCB), Agency for Science, Technology and Research (A*STAR), Singapore, Singapore. [7]Singapore Clinical Research Institute (SCRI), Singapore, Singapore. [8]Rheumatology and Immunology Service, KK Women's and Children's Hospital, Singapore, Singapore. [9]These authors contributed equally: Tira J. Tan, W. X. Gladys Ang, Who-Whong Wang. [10]These authors jointly supervised this work: Salvatore Albani, Han Chong Toh. ✉e-mail: salvo@duke-nus.edu.sg; toh.han.chong@singhealth.com.sg

