## [Peer Review File · Nature Communications]

A phase I study of an adenoviral vector delivering a MUC1/CD40-ligand fusion protein in patients with advanced adenocarcinomaReviewers' comments:

Reviewer #1 (Remarks to the Author): with expertise in cancer immunology – mass cytometry and network analysis

The article shows safety and some efficacy of a new CD40LMuc1 vaccine in a variety of relevant cancer patients.

The real strength of this article is the study of the immunome alongside the vaccine efficacy, and the characterisation of the immune response before and after treatment and in responders and non responders. Given that we have technology to study the immunome, I am very pleased to see this integrated into a clinical trial; it should be a responsibility of researchers to acquire as much data as possible from these patients. The data are sound and reporting of methods is generally well described. The supplementary data (other than heat maps, see below) are useful and relevant.

Major issue:

1. The use of tSNE plots to present the data is not very useful. It is very difficult to determine whether clusters are different or not, and the fold change calculations are not shown. tSNE are useful for determining what populations are present but not useful at all in determining differences between groups. There are several other platforms and visualisations available that allow statistical comparisons (eg CITRUS), direct overlays of groups (eg Vortex) and ways to identify different phenotypes (eg Brickplots). While the fold changes presented may seem different, the visualisation of tSNEs doesn't support convincing conclusions and a more robust demonstration of differences is needed. Similarly the heat maps in the supplementary data are overloaded with data and are not a helpful way to present the results to gain interpretation. The authors have amassed a lot of very good data, but have not presented it to its full advantage.

Minor issues:

1. please change title to "first-in-human". I assume many women would also like to be treated.
2. phrasing such as "trend towards" should be avoided, especially when differences have been defined as statistically significant or not.
3. please clarify the correlational network analysis. I found it difficult to interpret the figures (possibly too small) and link it to what was claimed in the text, this may be due to some unfamiliarity of the process, but other readers may benefit

Reviewer #2 (Remarks to the Author): with expertise in MUC1, cancer vaccines

This manuscript reports results from a Phase I trial of a vaccine composed of an adenovirus encoding MUC1 antigen fused to the extracellular domain of CD40L, administered to patients with several different advanced adenocarcinomas known to be MUC1+. The authors use state-of-the-art technology to analyze PBMCs pre and post vaccination.

This is a conceptually flawed manuscript where data do not support major conclusions. The claim is that the vaccine has changed the "immunome" in some patients and that this has led to a prolonged disease-free survival. Those patients who progressed slower are called "responders" and those who progressed faster are called "non-responders." This definition has nothing to do with a vaccine response. Even though MUC1-specific CTL were evaluated against 4 peptides from one of the MUC1 regions, these data were not used to determine responder vs non-responder groups. A responder to a vaccine is a person who makes vaccine

antigen-specific antibodies and T cells and a non-responder is a person who does not. Once those groups are determined, the clinical outcome can be correlated with that. The Kaplan-Meyer-like graph is completely wrong as "responders" and "non-responders" are not based on the vaccine response. The authors claim that the vaccines has changed the immunomes in responders, but in fact what they have looked at in great detail is a difference in the immunome between patients who progress faster and those who progress slower. Those differences may be interesting and important for disease outcome but they have nothing to do with the vaccine (or at least there is no support for that in the data).

Some specific problems are listed below:

1. The clinical trial is described in a schema but the vaccine vector is not and the references related to it are from 2004 and 2005. A big problem with vector-based MUC1 vaccines is that the vectors encode MUC1 production in normal cells that make normal and not tumor forms of the molecule. Normal MUC1 is not immunogenic.
2. The trial is too small for any sort of significant results. Only 14 patients, 3 different cancers, 5 different dose level groups.
3. Even if some changes were induced by the vaccine, it is hard to know if it was not all due to the CD40L and nothing to do with the antigen.

Reviewer #3 (Remarks to the Author): with expertise in MUC1 cancer vaccines - clinical trials

In this manuscript: "A first-in-man immunotherapy trial with a CD40L-MUC-1 vaccine induces clinically and mechanistically relevant remodeling of the immunome in epithelial malignancies overexpressing MUC1" Tan et al report a first-in-man Phase 1 study of Ad-sig-hMUC1/ecdCD40L (NCT02140996), an adenoviral vector vaccine encoding the ubiquitous tumor-associated antigen MUC1, with the extracellular domain of CD40 ligand as an adjuvant. The authors concluded that treatment was safe and tolerable in 21 patients with advanced carcinomas. They also did extensive Immune phenotyping of patients and identified possible differences between responders and non-responders.

Reviewer has the following comments:

1. Abstract is very generic and vague and should be more specific and reflect the manuscript
2. Sentence: "Cancer cells express antigens that differ qualitatively and/or quantitatively from those found on their normal counterparts – a premise which can be exploited to engender immunorecognition and rejection by the adaptive immune system" is not true, cancer cells also express antigens found on normal cells such as PSA, CEA, etc. Please modify accordingly.
3. Page 3 Safety, tolerability, and clinical responses to vaccination
 - a. Please describe grade 3 AEs.
 - b. New lesion is considered Progressive Disease per RECISTv 1.1, so please update accordingly
 - c. It is important to provide more data about study responders, which type of cancer they had, and were they taking other anticancer therapy (per-protocol Aromatase inhibitors and raloxifene were allowed.
 - d. We observed 29% (6/21) patients who had a time to next treatment > 180 days (Supplementary Figure 2), with significantly longer progression-free survival (Figure 1b) in responders compared to non-responders. How this was calculated, what was the p-value?
4. Page 3 Patient characteristics.
 - a. Please expand. How many sites?

- b. Please list main eligibility criteria.
- c. Was MUC1 expression evaluated on tumor biopsies?
- d. Provide more data about cancer subtypes: Breast ER/PR+, HER-2 status, Lung (non-small cell, small cell, adeno, squamous, mutational status). ovarian (serous, mucinous, etc). Please update Table 1 accordingly
5. Page 5. "We observed that the immunome of the cancer patients clustered differently from that of healthy controls (Figure 2a)" How healthy controls were selected? Did they sign the Informed Consent? Is that part of the clinical trial presented here?
6. Page 7. "Additionally, 29% of patients had a time to subsequent treatment of > 6 months, suggesting that vaccination had some degree of therapeutic efficacy in countering disease". Please provide more clinical data. Diagnosis, concurrent therapies, etc
7. Please specify how clinical responder's vs non responders were defined. Based on Table 3 it seems that all patients with stable disease were characterized as responders. Not sure one can call a patient with Stable Disease (and with tumor growth 10 – 15 %) a responder.
8. Per protocol Aromatase inhibitors and raloxifene were allowed. What is the impact of those drugs on immune system? How many responders and non-responders were taking those drugs. Please elaborate.
9. What was the MTD and recommended phase 2 dose?
10. What is the future plan for this vaccine?
11. Figure 3c. Correlational network analysis of CyTOF data in 10 patients before and after vaccination (correlation coefficient: 0.6). Please specify type of patients (responders, non - responders, cancer type, etc)

Reviewer #4 ((Remarks to the Author): with expertise in CD40L, clinical trials

The article is well written and presents interesting results in an important area. The most important comments concern clinical benefit. Otherwise I have only some moderate and several minor comments.

Abstract

Suggest changing "understanding on" to "understanding of"

It must be stated that no objective responses were achieved. Patients may experience stable disease as a reflection of the natural history of disease and not as a consequence of treatment.

Introduction

Compared to which treatments, please specify.

Please remove "3": ...cancer vaccination 3".

The authors should emphasize that the CD40L interaction they describe is not the only one: "...binds

CD40 receptors on dendritic cells..." Consider including the reference "Crossing the valley of death",

Results

Please add "locally": "...had **locally** recurrent or..."

Please add "locally": "the **locally** advanced or metastatic"

Please change "14 patients..." to "Fourteen patients..."

Discussion

"...of a prostate-specific..." to "...of a **prostate**-specific..."

"...due to their vaccination regimens and adjuvants..." Different patient groups is also an important difference that should be mentioned

"...immune checkpoint inhibitors and adoptive cell therapy..." I think that "adoptive cell therapy" should be removed since it is much less used

Please consider adding complement dependent cytotoxicity in "...for antibody-dependent cell-mediated cytotoxicity."

"The substantial proportion of patients..." is an exaggeration that must be changed since not a single objective response has been achieved in the study and if and to what extent the treatment has contributed to the stable diseases recorded is unknown

"...with immune checkpoint blockers (ICBs)..." Since these antibodies have been mentioned before, the abbreviation should be introduced earlier in the manuscript

Figure 1

Please change "followup" to "follow-up", this misspelling occurs twice.

Shouldn't 21 patients and not 19 be included in Figure 1b

It is very difficult to see the different colors (blue and red) in Figure 1b

Table 1

There are 22 instead of 21 patients listed under "ECOG performance status"

Supplementary Figure 2

Please change "pars" to "bars"

Supplementary Figure 3b

Please change "N" to "NR"

Supplementary Figure 5

Please explain "CYTOF" since a figure should be independent from the manuscript.

Supplementary Figures 8, 9 and 10

Please add when after vaccination the analyses were conducted

Reviewer #5 (Remarks to the Author): with expertise in biostatistics, clinical trial study design

This paper conducted a phase I clinical trial

1. I have concerns for the immortal person-time bias, especially Figure 1b for PFS between responders and non-responders. Time 0 (baseline) in this study is the start of the study, but responders and non-responders are defined post baseline, and therefore it's a problem to analyze them as they are baseline variables. Need to use either Landmark analysis or consider the response as a time-dependent covariate. Similarly, the Kaplan Meier curve is not appropriate for the difference defined by a time-dependent covariate.
2. The network analysis is basically drawing edges between two dots with high correlation, which implies those two dots are independent. It's possible the relationship between two dots is explained by their common relationship with other dots (hub dots) or other variables. The common method to identify edges is neighborhood selection (Meinshausen N, Bühlmann P, et al., 2006) or SPACE (Peng J, Wang P, Zhou N, et al.: Partial correlation estimation by joint sparse regression models. J Am Stat Assoc. 2009). By looking at the network plots between response and non-responder or before and after, it's not clear there is a difference.
3. The author claimed that "Our high-dimensional analysis evidenced key differences...". Though they didn't mention a high dimensional analysis method for the correlative study. The t-SNE plot is usually considered as "un-supervised" clustering analysis methods and not "high dimensional analysis".

Response to Reviewers

Reviewer #1

The article shows safety and some efficacy of a new CD40LMuc1 vaccine in a variety of relevant cancer patients.

The real strength of this article is the study of the immunome alongside the vaccine efficacy, and the characterisation of the immune response before and after treatment and in responders and non responders. Given that we have technology to study the immunome, I am very pleased to see this integrated into a clinical trial; it should be a responsibility of researchers to acquire as much data as possible from these patients. The data are sound and reporting of methods is generally well described. The supplementary data (other than heat maps, see below) are useful and relevant.

We thank the reviewer for their encouraging comments.

Major issue:

1. The use of tSNE plots to present the data is not very useful. It is very difficult to determine whether clusters are different or not, and the fold change calculations are not shown. tSNE are useful for determining what populations are present but not useful at all in determining differences between groups. There are several other platforms and visualisations available that allow statistical comparisons (eg CITRUS), direct overlays of groups (eg Vortex) and ways to identify different phenotypes (eg Brickplots). While the fold changes presented may seem different, the visualisation of tSNEs doesn't support convincing conclusions and a more robust demonstration of differences is needed. Similarly the heat maps in the supplementary data are overloaded with data and are not a helpful way to present the results to gain interpretation. The authors have amassed a lot of very good data, but have not presented it to its full advantage.

We thank the reviewer for pointing this out. We have now reorganised the data such that the clusters presented have both their phenotype and cluster frequencies reported clearly and directly, and avoided comparisons relying on tSNE visualisations. tSNE plots are now used only to visualise the global immunome. (Figure 1).

Minor issues:

1. please change title to "first-in-human". I assume many women would also like to be treated.

We thank the reviewer for this important recommendation and have made the appropriate changes to the title to be rightly inclusive.

2. phrasing such as "trend towards" should be avoided, especially when differences have been defined as statistically significant or not.

We have removed all such ambiguous phrasing.

3. please clarify the correlational network analysis. I found it difficult to interpret the figures (possibly too small) and link it to what was claimed in the text, this may be due to some unfamiliarity of the process, but other readers may benefit

We have reconsolidated and enlarged the network figures for improved clarity and isolated some nodes as examples (Figure 4, Supplementary Figures 7, 8). Inter-actable, enlargeable html files of Figures 4a and 4b have also been uploaded. We have also included an additional table listing and explaining the network properties discussed (Supplementary Table 8).

Reviewer #2

This manuscript reports results from a Phase I trial of a vaccine composed of an adenovirus encoding MUC1 antigen fused to the extracellular domain of CD40L, administered to patients with

several different advanced adenocarcinomas known to be MUC1+. The authors use state-of-the art technology to analyze PBMCs pre and post vaccination.

This is a conceptually flawed manuscript where data do not support major conclusions. The claim is that the vaccine has changed the "immunome" in some patients and that this has led to a prolonged disease-free survival. Those patients who progressed slower are called "responders" and those who progressed faster are called "non-responders." This definition has nothing to do with a vaccine response.

We thank the reviewer for the insightful comments, which we believe was eminently motivated by our previous poor definition of clinical responsiveness. We are truly thankful, again, for having raised this point as it has prompted us to revise our results in a more objective fashion. We have revised our comparisons in Figures 5 and 6 to include only samples classifiable by RECIST v. 1.1. Overall, the vaccine had objective, well documented effects on the immunome, as demonstrated by a multiplicity of supporting data, including the induction of MUC1-specific CTL (Supplementary Figure 2) as well as changes in the general architecture of the immunome (Figure 2, Figure 3, Figure 4).

Even though MUC1-specific CTL were evaluated against 4 peptides from one of the MUC1 regions, these data were not used to determine responder vs non-responder groups. A responder to a vaccine is a person who makes vaccine antigen-specific antibodies and T cells and a non-responder is a person who does not. Once those groups are determined, the clinical outcome can be correlated with that.

We appreciate this point but humbly beg to dissent, given the following considerations: (i) Antigen-specific CTL responses following therapeutic cancer vaccination do not consistently correlate with clinical responses nor overall survival in advanced solid tumour patients, as demonstrated in larger published cancer vaccine clinical trials including randomised Phase III trials and the cancer vaccine body of literature (1, 2). (ii) Importantly, the vaccine, which we employed, contains CD40 in addition to the MUC1 antigen and was specifically designed to broaden the immune response. Our ability to detect such level of "infectious immunogenicity" using a high dimensionality approach is actually the very essence of the paper.

The Kaplan-Meier-like graph is completely wrong as "responders" and "non-responders" are not based on the vaccine response. The authors claim that the vaccines has changed the immunomes in responders, but in fact what they have looked at in great detail is a difference in the immunome between patients who progress faster and those who progress slower. Those differences may be interesting and important for disease outcome but they have nothing to do with the vaccine (or at least there is no support for that in the data).

We thank the reviewer for pointing this out, and we have refrained from using Kaplan-Meier curves in this revised manuscript. Our analysis now focuses on the dysregulated systemic immunome in cancer patients and how this shifts in response to vaccination.

Some specific problems are listed below:

1. The clinical trial is described in a schema but the vaccine vector is not and the references related to it are from 2004 and 2005. A big problem with vector-based MUC1 vaccines is that the vectors encode MUC1 production in normal cells that make normal and not tumor forms of the molecule. Normal MUC1 is not immunogenic.

We have now included a diagram and provided additional details on the vaccine vector (Figure 1a). Due to the low immunogenicity of self-antigens such as MUC1, we engineered a vaccine construct that co-expresses it with the CD40L adjuvant. In a pre-clinical model, anergy to MUC1 has been successfully overcome by vaccination (3, 4).

2. The trial is too small for any sort of significant results. Only 14 patients, 3 different cancers, 5 different dose level groups.

We fully agree on this point with specific respect to clinical outcomes. We have indeed tuned down any reference to such effects. We accept that the number of patients in this study is small as a dose finding first-in-human Phase I clinical trial, and that a larger cohort is needed to support the

clinical and translational findings in this preliminary study. Indeed, the number of patients in our cohort was 21, with only 20 eligible for further downstream analyses. In some CyTOF analyses, additional samples were removed due to not meeting quality control standards, reducing the sample size further. However, with our application of deep analytic tools and platforms, we still observed statistically significant shifts in the immunome after vaccination despite this small number. We have clarified the number of patients used in each analysis in the figure legends.

3. Even if some changes were induced by the vaccine, it is hard to know if it was not all due to the CD40L and nothing to do with the antigen.

We agree with the reviewer that we are unable to determine and separate the effect of CD40L adjuvant from cancer antigen (MUC-1) in this Phase I trial. In pre-clinical studies, local immunotherapy with AdCD40L increased cytotoxic CD8 T cells (5) and promoted DC maturation (6), which aligns with the observations in our manuscript. However, we argue that our goal is to document broad, distinct immune shifts and phenomena induced by this novel vaccine construct in its multi-component entirety exerting its intended clinical effect. This is the first application of such an immunome approach in any therapeutic cancer vaccine clinical study. Certainly more such studies are required to ascertain immunome effects whether against a sole cancer-specific antigen vaccine or in a multi-component construct. Similarly, further studies will be required to determine the relative contribution of CD40L as part of a cancer antigen targeting cancer vaccine construct.

Reviewer #3 (Remarks to the Author): with expertise in MUC1 cancer vaccines - clinical trials

In this manuscript: "A first-in-man immunotherapy trial with a CD40L-MUC-1 vaccine induces clinically and mechanistically relevant remodeling of the immunome in epithelial malignancies overexpressing MUC1" Tan et al report a first-in-man Phase 1 study of Ad-sig-hMUC1/ecdCD40L (NCT02140996), an adenoviral vector vaccine encoding the ubiquitous tumor-associated antigen MUC1, with the extracellular domain of CD40 ligand as an adjuvant. The authors concluded that treatment was safe and tolerable in 21 patients with advanced carcinomas. They also did extensive Immune phenotyping of patients and identified possible differences between responders and non-responders.

We thank the reviewer for their thoughtful comments.

Reviewer has the following comments:

1. Abstract is very generic and vague and should be more specific and reflect the manuscript

We have revised the abstract to be more specific to reflect the results obtained in the manuscript.

2. Sentence: "Cancer cells express antigens that differ qualitatively and/or quantitatively from those found on their normal counterparts – a premise which can be exploited to engender immunorecognition and rejection by the adaptive immune system" is not true, cancer cells also express antigens found on normal cells such as PSA, CEA, etc. Please modify accordingly.

We have replaced the sentence as appropriate.

3. Page 3 Safety, tolerability, and clinical responses to vaccination

a. Please describe grade 3 AEs.

b. New lesion is considered Progressive Disease per RECISTv 1.1, so please update accordingly

c. It is important to provide more data about study responders, which type of cancer they had, and were they taking other anticancer therapy (per-protocol Aromatase inhibitors and raloxifene were allowed).

a. Grade 3 AEs are now described in Supplementary Table 2.

b. We have updated the tumour response classifications in Supplementary Table 3.

c. We have provided more data regarding the study participants in Supplementary Table 1 which includes types of cancer as well as relevant subtypes, the number of lines of prior systemic therapy, and concurrent endocrine therapies.

d. We observed 29% (6/21) patients who had a time to next treatment > 180 days (Supplementary Figure 2), with significantly longer progression-free survival (Figure 1b) in responders compared to non-responders. How this was calculated, what was the p-value?

The time to next treatment was calculated as time from first study vaccine injection to date of documented start of next line of therapy. We apologise for the confusion caused with a following comment on progression free survival comparison between our dichotomised cohort of “responders” vs “non-responders”. Progression free survival was taken as time from first study vaccine injection to first documented clinical or radiological progression of disease. This has been removed from the report. P-value was not reported due to small sample size and significance in progression free survival should not be claimed.

4. Page 3 Patient characteristics.

a. Please expand. How many sites?

b. Please list main eligibility criteria.

c. Was MUC1 expression evaluated on tumor biopsies?

d. Provide more data about cancer subtypes: Breast ER/PR+, HER-2 status, Lung (non-small cell, small cell, adeno, squamous, mutational status). ovarian (serous, mucinous, etc). Please update Table 1 accordingly

a. This was a single-site study.

b. The eligibility criteria was recurrent or metastatic adenocarcinomas of the breast, ovary, or lung as proven by biopsy, and/or elevated MUC1 levels determined by tumour immunohistochemistry or serum tumour marker.

c. Please see (b)

d. Additional information regarding cancer subtypes is now provided in Supplementary Table 1.

5. Page 5. “We observed that the immunome of the cancer patients clustered differently from that of healthy controls (Figure 2a)” How healthy controls were selected? Did they sign the Informed Consent? Is that part of the clinical trial presented here?

Healthy controls were not part of this clinical trial. Samples were collected by KK Women's and Children's Hospital Singapore with ethics approval from the SingHealth Centralised Institutional Review Board and after informed consent was obtained. This cohort was previously described (7, 8).

6. Page 7. “Additionally, 29% of patients had a time to subsequent treatment of > 6 months, suggesting that vaccination had some degree of therapeutic efficacy in countering disease”. Please provide more clinical data. Diagnosis, concurrent therapies, etc

Please refer to Supplementary Table 1 for more information on subject profiles. In particular, we find it notable that certain patients, such as A14, an endometrioid ovarian cancer patient with two prior lines of therapy, still saw tumour shrinkage (-25.8% in this patient) despite their disease outlook.

7. Please specify how clinical responder’s vs non responders were defined. Based on Table 3 it seems that all patients with stable disease were characterized as responders. Not sure one can call a patient with Stable Disease (and with tumor growth 10 – 15 %) a responder.

We have revised our patient classification based on feedback (Supplementary Table 3). Briefly, we have strictly restricted our stratified analyses to RECIST classifiable samples (stable disease and progressive disease). We apologise for the confusion.

8. Per protocol Aromatase inhibitors and raloxifene were allowed. What is the impact of those drugs on immune system? How many responders and non-responders were taking those drugs. Please elaborate.

We thank the reviewer for the helpful comment on immunomodulatory effects of endocrine therapy, which are likely to be complex. Indeed, a recent study suggests that estrogen deprivation enhances PD-L1 expression and immunogenicity (9). Estrogen has known immune regulatory properties, which may enhance immune responses upon inhibition. For example, a reduction in FoxP3+ T cells(10) and an increase in CD8 T cell infiltration(11) have been seen following letrozole / exemestane treatment.

3 subjects with stable disease (SD), 1 with progressive disease (PD), and 1 with non-evaluable (NE) disease were on aromatase inhibitors (total n=5). No subjects on study were treated with raloxifene.

1 SD patient had 8 prior lines of endocrine therapy with re-challenge of letrozole started just prior to 1st vaccine injection. 2 subjects (1 SD and 1 NE) had been on letrozole therapy > 6 months prior to 1st vaccine injection. 1 SD subject commenced exemestane 1 month prior to 1st vaccine. The PD subject on aromatase inhibitors had 2 prior lines of endocrine therapy and was started on exemestane just prior to vaccination.

9. What was the MTD and recommended phase 2 dose?

The maximally administered dose was 10¹¹ viral particles, with dose levels 5-7 evaluating impact of multiple vaccinations. We were unable to determine the MTD as all doses tested were safe and well-tolerated. Future studies are required to determine the optimal dose and schedule of this vaccine.

10. What is the future plan for this vaccine?

Based on the safety profile in this first-in-human Phase 1 clinical trial, this MUC-1-CD40L vaccine has been planned to start in Europe as a Phase II clinical trial in a larger cohort across selected solid tumours in combination with an immune checkpoint inhibitor.

11. Figure 3c. Correlational network analysis of CyTOF data in 10 patients before and after vaccination (correlation coefficient: 0.6). Please specify type of patients (responders, non-responders, cancer type, etc)

The type of patients are now listed in Supplementary Table 7.

Reviewer #4 ((Remarks to the Author): with expertise in CD40L, clinical trials (see document attached)

The article is well written and presents interesting results in an important area. The most important comments concern clinical benefit. Otherwise I have only some moderate and several minor comments.

We thank the reviewer for their positive comments.

Abstract

Suggest changing “understanding on” to “understanding of”

It must be stated that no objective responses were achieved. Patients may experience stable disease as a reflection of the natural history of disease and not as a consequence of treatment.

We have made the corrections as suggested and made note of the important caveats.

Introduction

Compared to which treatments, please specify.

Please remove “3”: ...cancer vaccination 3”.

The authors should emphasize that the CD40L interaction they describe is not the only one:

“...binds CD40 receptors on dendritic cells...” Consider including the reference “Crossing the valley of death”,

We have made the corrections accordingly and added the suggested reference.

Results

Please add “locally”: “...had locally recurrent or...

Please add “locally”: “the locally advanced or metastatic”

Please change “14 patients...” to “Fourteen patients...”

We have amended the wording for clarity.

Discussion

“...of a prostrate-specific...” to “...of a prostate-specific...”

“...due to their vaccination regimens and adjuvants...” Different patient groups is also an important difference that should be mentioned

“...immune checkpoint inhibitors and adoptive cell therapy...” I think that “adoptive cell therapy” should be removed since it is much less used

Please consider adding complement dependent cytotoxicity in “...for antibody-dependent cell-mediated cytotoxicity.”

We have made the corrections accordingly and added the suggested mechanism associated references.

“The substantial proportion of patients...” is an exaggeration that must be changed since not a single objective response has been achieved in the study and if and to what extent the treatment has contributed to the stable diseases recorded is unknown

We agree with this statement and have changed our wording to be more accurate.

“...with immune checkpoint blockers (ICBs)...” Since these antibodies have been mentioned before, the abbreviation should be introduced earlier in the manuscript

We have made the above changes accordingly.

Figure 1

Please change “followup” to “follow-up”, this misspelling occurs twice. Shouldn't 21 patients and not 19 be included in Figure 1b?

We have corrected the misspelling as well as the error in patient numbers.

It is very difficult to see the different colors (blue and red) in Figure 1b

We have changed the colour scheme for better visualisation.

Table 1

There are 22 instead of 21 patients listed under “ECOG performance status”

We have corrected the error.

Supplementary Figure 2

Please change “pars” to “bars”

We have made the correction .

Supplementary Figure 3b

Please change “N” to “NR”

We have corrected and revised our classifications of response (Supplementary Table 3) and have ensured correct labelling.

Supplementary Figure 5

Please explain "CYTOF" since a figure should be independent from the manuscript.

We have changed the term to "mass cytometry" for greater clarity.

Supplementary Figures 8, 9 and 10

Please add when after vaccination the analyses were conducted

We have added the information to the Methods section.

Reviewer #5 (Remarks to the Author): with expertise in biostatistics, clinical trial study design

This paper conducted a phase I clinical trial

1. I have concerns for the immortal person-time bias, especially Figure 1b for PFS between responders and non-responders. Time 0 (baseline) in this study is the start of the study, but responders and non-responders are defined post baseline, and therefore it's a problem to analyze them as they are baseline variables. Need to use either Landmark analysis or consider the response as a time-dependent covariate. Similarly, the Kaplan Meier curve is not appropriate for the difference defined by a time-dependent covariate.

We thank the reviewer for pointing this out. We agree with the reviewer's comment and have removed PFS analyses for the different patient groups.

2. The network analysis is basically drawing edges between two dots with high correlation, which implies those two dots are independent. It's possible the relationship between two dots is explained by their common relationship with other dots (hub dots) or other variables. The common method to identify edges is neighborhood selection (Meinshausen N, Bühlmann P, et al., 2006) or SPACE (Peng J, Wang P, Zhou N, et al.: Partial correlation estimation by joint sparse regression models. J Am Stat Assoc. 2009). By looking at the network plots between response and non-responder or before and after, it's not clear there is a difference.

We modelled a correlation network as a way of inferring direct or indirect interaction between cell subsets. The underlying reasoning behind this approach is that if two cell subsets move in the same or opposite directions, they may be influencing or regulating each other via direct or indirect interactions. Hence, we used high correlation to establish the edge (interaction) between two subsets. We accept that there are other methods of inferring interactions between subsets that we have not used here. Correlation networks have previously been used extensively in gene regulatory network analysis to find meaningful biological information. With regards to our network plots, we have revised them for greater clarity and added examples of certain nodes and their connections to highlight differences between groups.

3. The author claimed that "Our high-dimensional analysis evidenced key differences...". Though they didn't mention a high dimensional analysis method for the correlative study. The t-SNE plot is usually considered as "un-supervised" clustering analysis methods and not "high dimensional analysis".

We apologise for the confusion. We would like to clarify that the high-dimensional aspect of our analysis is the 71 unique markers we have used for mass cytometry. Our pipeline then uses unsupervised clustering by FlowSOM and subsequent dimensional reduction and visualisation.

References

1. G. Parmiani *et al.*, Cancer immunotherapy with peptide-based vaccines: what have we achieved? Where are we going? *J Natl Cancer Inst* **94**, 805-818 (2002).
2. S. L. Slezak, A. Worschech, E. Wang, D. F. Stroncek, F. M. Marincola, Analysis of vaccine-induced T cells in humans with cancer. *Adv Exp Med Biol* **684**, 178-188 (2010).
3. L. Zhang *et al.*, An adenoviral vector cancer vaccine that delivers a tumor-associated antigen/CD40-ligand fusion protein to dendritic cells. *Proc Natl Acad Sci U S A* **100**, 15101-15106 (2003).
4. Y. Tang *et al.*, Multistep process through which adenoviral vector vaccine overcomes anergy to tumor-associated antigens. *Blood* **104**, 2704-2713 (2004).
5. C. Lindqvist, L. C. Sandin, M. Fransson, A. Loskog, Local AdCD40L gene therapy is effective for disseminated murine experimental cancer by breaking T-cell tolerance and inducing tumor cell growth inhibition. *J Immunother* **32**, 785-792 (2009).
6. A. S. Loskog, M. E. Fransson, T. T. Totterman, AdCD40L gene therapy counteracts T regulatory cells and cures aggressive tumors in an orthotopic bladder cancer model. *Clin Cancer Res* **11**, 8816-8821 (2005).
7. J. G. Yeo *et al.*, The Extended Polydimensional Immunome Characterization (EPIC) web-based reference and discovery tool for cytometry data. *Nat Biotechnol* **38**, 679-684 (2020).
8. J. G. Yeo *et al.*, A Virus-Specific Immune Rheostat in the Immunome of Patients Recovering From Mild COVID-19. *Front Immunol* **12**, 674279 (2021).
9. D. Huhn *et al.*, Prolonged estrogen deprivation triggers a broad immunosuppressive phenotype in breast cancer cells. *Mol Oncol* **16**, 148-165 (2022).
10. D. Generali *et al.*, Immunomodulation of FOXP3+ regulatory T cells by the aromatase inhibitor letrozole in breast cancer patients. *Clin Cancer Res* **15**, 1046-1051 (2009).
11. M. S. Chan *et al.*, Changes of tumor infiltrating lymphocyte subtypes before and after neoadjuvant endocrine therapy in estrogen receptor-positive breast cancer patients--an immunohistochemical study of Cd8+ and Foxp3+ using double immunostaining with correlation to the pathobiological response of the patients. *Int J Biol Markers* **27**, e295-304 (2012).

REVIEWERS' COMMENTS

Reviewer #1 (Remarks to the Author):

The revised study addresses many of the issues raised in my initial review. However, some additional comments:

1. The abstract remains vague and confusing. I suggest the authors outline their proposed endpoints more clearly, ie 1) efficacy of vaccine 2) opportunistic study of immunome in vaccinated populations
2. please check for remaining "in-man" phrasing
3. Results section titles could be more informative - it is difficult to follow which groups are being compared and why.
4. While I understand the difficulty in recruiting patients, the variability between individuals and cancer types confounds these results. I have no problem with small group studies looking at efficacy or small changes. However, high dimensional cytometry comes with significant noise. The authors validate their findings with manual gating, which is good, but the actual results are a little underwhelming. The main differences are in cells with very broad phenotypes (eg B cells or CD45+ CD4+ cells) or cells already well defined in cancer (effector CD8+ cells). As such, I am not sure of the novelty of the data and its contribution to advancing research.
5. I am very pleased that the study if the immunome is included in clinical trials and I commend the authors for taking this approach.
6. Figure 5 - the data are presented from SD patients, PD patients could be compared in this figure too.

Reviewer #2 (Remarks to the Author):

I appreciate the effort of the authors to reply thoughtfully to my comments and make several revisions based on them.

I still strongly disagree with their rebuttals and the claim of both the original and the revised manuscript that they are reporting the change in the immunome as a result of the vaccine. All the comparisons are still only between people who progressed versus those who did not, which to me, in the absence of non-vaccinated controls, reflects their immunome differences and reaction to disease progression, independent of vaccination.

I do not agree with all the hand-waving about the immune response not correlating with clinical outcome. If a strong enough response were to be elicited, there would be a correlation. I assume that the authors created the MUC1-CD40 vector to improve the response, but judging by antigen-specific responses they are measuring, this did not happen.

In summary, this is not a cancer vaccine paper. The data may still be very useful but it needs to be packaged in a different manuscript that does not claim vaccine-induced immunome alterations.

Reviewer #3 (Remarks to the Author):

The authors' responses and revisions have satisfactorily addressed my comments on the earlier version of the manuscript. I have no additional comments or suggestions.

Reviewer #4 (Remarks to the Author):

I am happy with the authors responses to my questions and the actions they have taken. Hence, I recommend publication.

Reviewer #5 (Remarks to the Author):

The authors removed the inappropriate plots and addressed the concerns. Looks good.

Reviewer's Comments:

Reviewer #1 (Remarks to the Author)

The revised study addresses many of the issues raised in my initial review. However, some additional comments:

1. The abstract remains vague and confusing. I suggest the authors outline their proposed endpoints more clearly, ie 1) efficacy of vaccine 2) opportunistic study of immunome in vaccinated populations
2. please check for remaining "in-man" phrasing
3. Results section titles could be more informative - it is difficult to follow which groups are being compared and why.
4. While I understand the difficulty in recruiting patients, the variability between individuals and cancer types confounds these results. I have no problem with small group studies looking at efficacy or small changes. However, high dimensional cytometry comes with significant noise. The authors validate their findings with manual gating, which is good, but the actual results are a little underwhelming. The main differences are in cells with very broad phenotypes (eg B cells or CD45+ CD4+ cells) or cells already well defined in cancer (effector CD8+ cells). As such, I am not sure of the novelty of the data and its contribution to advancing research.
5. I am very pleased that the study if the immunome is included in clinical trials and I commend the authors for taking this approach.
6. Figure 5 - the data are presented from SD patients, PD patients could be compared in this figure too.

We thank the reviewer for their insightful comments and suggestions.

1. We have revised the abstract further to reflect the study's proposed endpoints.
2. We have checked the manuscript and changed "in-man" to "in-human".
3. We have made the results section titles more informative and less vague to reflect to comparisons between groups and results obtained.
4. We are very thankful to this review for giving the opportunity to further clarify this point. We believe that a fundamental level of novelty in this context is to assess changes found before and after vaccination in the overall architecture of Immunome and, importantly, of the relational networks it is based on. This finding underscores the complexity of the effects of the treatment, which only a high dimensionality approach like the one, which we propose, can identify. In this perspective, the differences are actually striking. The individual signatures, which we distilled from this analysis for their potentially predictive value are few and broad, which, in our opinion, is consistent with major immune functions which need to be meaningful to be relevant. These signatures, as we clearly state in the manuscript, have to be validated in an independent study.
5. We thank the reviewer for their comments.
6. While we have compared PD patients, the small number failed to turn out any significant hits. We have included some of the more interesting findings with large magnitude changes, but did not make the $p < 0.05$ cut-off, in Supplementary Figure 10.

Reviewer #2 (Remarks to the Author)

I appreciate the effort of the authors to reply thoughtfully to my comments and make several revisions based on them.

We thank the reviewer for their many insightful suggestions and comments, which have certainly helped improving the manuscript and its clarity.

I still strongly disagree with their rebuttals and the claim of both the original and the revised manuscript that they are reporting the change in the immunome as a result of the vaccine. All the comparisons are still only between people who progressed versus those who did not, which to me, in the absence of non-vaccinated controls, reflects their immunome differences and reaction to disease progression, independent of vaccination.

I do not agree with all the hand-waving about the immune response not correlating with clinical outcome. If a strong enough response were to be elicited, there would be a correlation. I assume that the authors created the MUC1-CD40 vector to improve the response, but judging by antigen-specific responses they are measuring, this did not happen.

In summary, this is not a cancer vaccine paper. The data may still be very useful but it needs to be packaged in a different manuscript that does not claim vaccine-induced immunome alterations.

We respectfully disagree with the assertion that comparisons are only between progressors and non-progressors. We have compared baseline and post-vaccination samples in the cohort before stratification into stable or progressive disease (Figure 2). While we acknowledge that our study was not specifically designed to look at efficacy, and thus lacked a control cohort, the patients had a highly heterogeneous clinical history (mix of cancer types, different number of prior treatments etc.) (Supplementary Table 1). We also demonstrate in Supplementary Figures 4 and 5 that the baseline immunome is highly compromised, and thus, any observed changes post-vaccination would be notable. In our interpretation, the fact that we could observe statistically significant differences between pre- and post-vaccine conditions given the heterogeneous cohort strongly suggests a vaccine-mediated, cause and effect relationship. Particularly in the last version of the manuscript, we are as careful as possible to not overstate any of these effects. With respect to the issue of correlation of ELISpot results with clinical outcomes, we acknowledge in the manuscript: "MUC1-specific IFN γ responses were significantly induced by vaccination (Supplementary Figure 2a). This occurred despite pre-existing seropositivity to adenovirus (AdV) present in the majority (15/21) of patients prior to vaccination and the elevation of anti-AdV titres post-vaccination (Supplementary Figure 2b). However, ELISpot has significant limitations, as: i) it has been reported in the literature that antigen-specific responses to cancer vaccines commonly do not correlate with clinical outcome^{1, 2, 3} and ii) it does not have the sensitivity or dimensionality to take into account other broader immune responses in response to the intervention, such as antigenic cascade⁴, which may lead to changes in the overall immunome of patients. This led us to investigate immune changes elicited by Ad-sig-hMUC1/ecdCD40L in a more holistic manner."

Reviewer #3 (Remarks to the Author)

The authors' responses and revisions have satisfactorily addressed my comments on the earlier version of the manuscript. I have no additional comments or suggestions.

We thank the reviewer for their helpful comments and for taking the time to review the manuscript again.

Reviewer #4 (Remarks to the Author)

I am happy with the authors responses to my questions and the actions they have taken. Hence, I recommend publication.

We thank the reviewer for their helpful comments and efforts in reviewing the manuscript again.

Reviewer #5 (Remarks to the Author)

The authors removed the inappropriate plots and addressed the concerns. Looks good.

We thank the reviewer for their helpful comments and for their time in reviewing the manuscript again.

References

1. Lee KH, *et al.* Increased vaccine-specific T cell frequency after peptide-based vaccination correlates with increased susceptibility to in vitro stimulation but does not lead to tumor regression. *J Immunol* **163**, 6292-6300 (1999).
2. Lyerly HK. Quantitating cellular immune responses to cancer vaccines. *Semin Oncol* **30**, 9-16 (2003).
3. Rosenberg SA, Yang JC, Restifo NP. Cancer immunotherapy: moving beyond current vaccines. *Nat Med* **10**, 909-915 (2004).
4. Gulley JL, *et al.* Role of Antigen Spread and Distinctive Characteristics of Immunotherapy in Cancer Treatment. *J Natl Cancer Inst* **109**, (2017).